# Quality of Leadership and Organizational Climate in a Sample of Spanish Workers. The Moderation and Mediation Effect of Recognition and Teamwork

**DOI:** 10.3390/ijerph17010032

**Published:** 2019-12-18

**Authors:** Carlos Pérez-Vallejo, Juan José Fernández-Muñoz

**Affiliations:** 1International Doctoral School Health Sciences, University Rey Juan Carlos, 28933 Alcorcón, Madrid, Spain; 2Department Postgraduate study University Francisco de Vitoria, 28224 Pozuelo de Alarcón, Madrid, Spain; 3Area of Behavioral Sciences Methodology, Department of Psychology, University Rey Juan Carlos, 28933 Alcorcón, Madrid, Spain; juanjose.fernandez@urjc.es

**Keywords:** moderated mediation, leadership, organizational climate, recognition, teamwork

## Abstract

The purpose of this study is to analyze the relationships between the quality of leadership, achievement recognition, and teamwork with the organizational climate and quality of life at work. A questionnaire was prepared that included all items of the variables in this study of the scales ECO IV and ISTAS21. The sample selected was composed by 1179 workers of a multinational company; mediation and moderation analysis was applied with Process v3.4. The results of this study suggest that teamwork exerts significantly the expected mediating effect in the relationship between the quality of leadership and the organizational climate. However, recognition of achievement does not produce moderation in the relationship between leadership quality with the organizational climate. To sum up, leadership quality, teamwork, and recognition of achievements improve the perception of the organizational climate and quality of life at work. Therefore, the organization must establish its own leadership style that allows it to achieve its objectives and improve the quality of life of workers.

## 1. Introduction

The impact of leadership on the organizational climate is discussed in the literature. However, further insight is necessary into how to improve the organizational climate through leadership based on recognition of achievement and teamwork.

Research has always studied the role of leadership in improving the effectiveness of employees and the subsequent effect on the results of the organization. Currently, interest in analyzing the role of leadership in the health and welfare of workers has increased. [1].

Throughout history, different theories have been established that strive to explain the figure of the leader and his relationship with the different actors in the context; studying his evolution throughout history to better understand him [2].

The concept of leadership is complex and has been studied from multiple perspectives and approaches, which is constantly evolving and updating [1]. We can find studies focused on the characteristics, attributes, and personal traits of the leader himself [3]; other studies such as those of the Ohio State University [4] and the University of Michigan [5], related to behavior, influence or leadership styles; others such as contingency theories, study leadership regarding the environment or the organization and consider the tasks and relationships with the followers [6,7]; and there are new perspectives that explain the effects of the relations of influence and exchange between leaders and followers. [8]

We understand in this study, the quality of leadership as the behaviours and attributes of the head or direct supervisor that allow us to judge their value as a leader [9].

Good leadership must have the capacity to resolve conflicts, plan and distribute work equally, show concern for the welfare of their subordinates, and have effective communication skills [9].

The quality of leadership in the management of human teams carried out by the immediate supervisors is related to the social support of superiors and their capacity in the application of personnel management procedures [9]. Leadership, as a basic research topic in organizational psychology, has been justified by its importance for the functioning and success of organizations. In the extent to which theories about leadership are developed, it will improve the techniques of selection and training of leaders, to increase organizational effectiveness [10].

How the worker perceives that his immediate superior behaves and how the company treats him, are two aspects to consider if one wants to analyze the perception of the work environment, so establishing a leadership style and developing appropriate human resources practices are of higher priority than other organizational aspects [11]. If we consider this, the applied leadership style directly affects the organization and influences the behavior of the workers [12].

Organizations require transformative leaders who achieve the objectives that generate a good organizational climate [11].

Organizations with positive organizational climates have greater adaptability and they face uncertainty better, favoring innovation and development as an organization [12]. Numerous researches have also shown that the organizational climate affects job satisfaction and motivation at work [13].

The organizational climate is one of the most outstanding indicators regarding the quality of work-life, because it provides a faithful and specific approximation of the perceptions that the workers of the organization have and of the different realities of work [12].

Several studies on organizational climate indicate that social interaction between members of the work team and leader-member interaction are factors that promote the improvement of the organizational climate [14].

Teamwork is the degree to which this organizational mode is perceived to exist in the company and is convenient for both the employee and the company [13].

In this sense, [15] the definition of a team is a group of people with complementary skills, committed to a common purpose, with a set of performance goals, and with an approach for which they feel mutually responsible.

According to these authors [15], groups that fail in their attempt to become teams rarely have in common specific objectives that lead them to the development of actions due to a lack of effort, poor leadership, insufficient focus on the objectives, and no concurrence in aspirations.

As some authors have related teamwork with worker satisfaction [16], others indicated that in certain circumstances, teamwork can also generate dissatisfaction among team members [17]. This indicates that teamwork is a double-edged tool, in which depending on the organizational context, the members, the activity, and even the leadership exercised on the team, can generate satisfaction or dissatisfaction in the workers and, therefore, both can affect the quality of life at work. But what usually comes together is that the climate of the work teams can have an important influence on the team’s results [18].

An essential role of leadership is to recognize the contribution of workers and their performance to the achievement of objectives. When workers are recognized for their good work, the actions and behaviors desired by the organization coincide with the culture and objectives established in the organization being reinforced, thus achieving greater motivation in the worker [19].

The esteem or recognition of achievement includes the recognition of managers to the effort made to perform the work, to receive adequate support, and fair treatment at work. The esteem represents a psychological compensation obtained sufficiently or insufficiently in exchange for the work performed and together with the prospects for promotion, job security, working conditions, and a salary appropriate to the demands of the job, constitutes the basis of compensation [9]. 

### Hypothesis Development

In one study made by [20] involving about 99 research studies that examined the effect of the impact of leadership on the organizational climate, the results of the meta-analysis found a large positive effect of leadership on the organizational climate. This finding suggests that a leader has an important place in the establishment of the organizational climate and that there is a strong relationship between leadership and organizational climate. Therefore, leaders have an important role to play in ensuring the productivity and sustainability of an organization and in the establishment of a positive organizational climate [21].

An effective leadership style can improve the organizational climate and, therefore, develop the organization’s morale and upturn employee effectiveness [21]. The findings of these authors’ study [20], reconfirm the previous studies’ outcomes that are associated with the positive influence of leadership style on the organizational climate.

According to the “effort-reward model” of [22], the interaction between high effort and low level long term rewards represents the situation of the greatest health risk. The applied style of leadership, recognition of achievement in relation to effort, and promoting teamwork, directly affect the climate in the organization and influence the behavior of its members.

Balancing leadership and teamwork as two sides of the same coin has become necessary in an increasingly competitive environment, especially in the face of scarcity of valuable resources. To take advantage of the best of a team in the relentless pursuit of organizational success requires a greater emphasis on social leadership skills. The leader has an essential role in generating a positive organizational climate in the work team [23].

Recognition of achievement of leadership is a particular characterization of leader behavior that emphasizes commitment to developing personal relationships with followers, care and concern for others, willingness to attend to the unique preferences and work styles of subordinates, and facilitating cooperation among members of a work group [24] When employees are recognized they feel more appreciated in their jobs and acknowledged by their managers.

This study aims to determine the extent to which leadership style based on recognition of achievement and teamwork has a significant influence on the organizational climate as a positive aspect for improving the quality of life at work [13]. For this, the following hypotheses have been raised:

**Hypothesis 1 (H1).** 
*The direct effects of leadership on the organizational climate are mediated by teamwork and, in turn, moderated by the recognition of the achievement of that leadership, so that:*


**Hypothesis 1a (H1a).** 
*Teamwork mediates the relationship between the leadership predictor variable and the organizational climate criteria variable.*


**Hypothesis 1b (H1b).** 
*The recognition of leadership achievement moderates the relationship between the leadership predictor variable and the organizational climate criteria variable.*


## 2. Material and Methods

### 2.1. Participants and Procedure

The sample selected was composed of 1179 workers of a multinational energy services company in process digital and technological transformation. Of them, 40.7% worked in Madrid, 20.2% in Barcelona, and the rest (39.1%) are placed in several cities, e.g., Valencia, Seville, Bilbao, Malaga, Zaragoza or Valladolid. Of the workers, 81.1% were male and 18.9% were female. The distribution of their age included 2.7% under 26 years, 17.8% between 26 and 35 years, 28.3% between 36 and 45 years, 31.1% between 46 and 55 years, and 10.1% over 55 years.

In terms of labor context, 74.6% had an open-ended contract, while 25.4% had a temporary contract. The distribution of length of service was: 26.9% less than two years, 24.8% between two and ten years, 34.5% between ten and twenty years, and 13.8% more than twenty years. Finally, 42.6% were blue-collar workers, 10.9% were administrative, 22.5% were qualified workforce, 23.3% were white-collar workers, and 0.7% were managers.

This study had a cross-sectional design. The data were recollected through a convenience sampling and the tool was a questionnaire. The participants received an online invitation to participate in this study (email with a google docs link to complete the questionnaire online); moreover, they were also informed of the objectives and the anonymity of the data collected. There were no exclusion criteria. These data were collected between May 2018 and July 2018.

### 2.2. Instruments

The questionnaire used in this study had several sections. Here, we have included the scales used to check the variables. Prior to the final questionnaire, a pilot study was applied with a sample of 50 workers from the multinational company.

Recognition was measured with the ISTAS21 (2004). This instrument is a Spanish adaptation of the Copenhagen Psychosocial Questionnaire—CoPsoQ [9]. The number of items is 4 with a Likert response from 0 to 4 where 0 = never and 4 = always. Therefore, a higher score indicates a better perception of the recognition received. The internal consistency (Cronbach’s Alpha) was α = 0.74.

Quality of leadership was measured with the ISTAS21 (2004). This instrument is a Spanish adaptation of the Copenhagen Psychosocial Questionnaire—CoPsoQ [9]. The number of items is 4 with a Likert scale from 0 to 4. Therefore, a higher score indicates a better perception of the quality of leadership. The internal consistency (Cronbach’s Alpha) was α = 0.90.

Organizational climate was measured with the instrument ECO IV [13]. This diagnostic instrument measures the organizational climate through 10 dimensions: 5 management (organizational clarity, coherence, availability of resources, stability, and compensation); 3 collective (teamwork, manager’s support, collective values); and 2 individual (a feeling of belonging and engagement). The number of items is 46 with a Likert response from 0 to 6, a higher score indicates a better perception of the organizational climate. The internal consistency (Cronbach’s Alpha) was α = 0.94.

Teamwork was measured with the instrument ECO IV [13]. The number of items is 4 with a Likert scale from 0 to 6, a higher score indicates a better perception of the teamwork. The internal consistency (Cronbach´s Alpha) was α = 0.73.

### 2.3. Analysis

The statistical analyses were applied with SPSS 25.0 and with PROCESS 2.16.3 [25]. Descriptive analysis and Pearson correlations for all variables were also included in the results section. Assumptions were checked to ensure the application of the linear regression model and mediation and moderation model, for instance: normality *p* > 0.05 (Kolmogorov test), homoscedasticity *p* < 0.05 (Levene test), and independency of the variables D-W = 1.95 (Durbin-Watson). For the moderation and mediation relationships, model 5 according to Hayes (2013) was applied with a 95% CI and a bootstrapping of 10,000. In order to support the significant effect of the variables inside of the model p values < 0.01 and LLCI and ULCI was included in the model. Moreover, pick a point and the Johnson-Neyman test was applied in order to know the effects of the independent variable on the dependent variable of the model [26].

## 3. Results

### 3.1. Descriptive Findings and Correlation

Table 1 shows the descriptive findings and the significant correlation with a confidence level of *p* < 0.01 and 0.05 between the variables included in the mediation model: organizational climate, quality of leadership, recognition, and teamwork. Moreover, Cronbach´s alpha of these dimensions was included in the table. These alphas were between 0.74 and 0.94. The correlations were significant between all dimensions, for instance: organizational climate and quality of leadership (r = 0.830, *p* < 0.01) or organizational climate and teamwork (r = 0.779, *p* < 0.01).

### 3.2. Moderated-Mediation Model

Table 2 shows the findings of the moderated-mediation model. In this sense, there was a significant direct effect of leadership on organizational climate (B = 0.224, *p* < 0.01, 95% CI: 0.18 to 0.26) and also a significant direct effect of teamwork on organizational climate (B = 0.253, *p* < 0.01, 95% CI: 0.23 to 0.28). Moreover, the indirect effects of the quality of leadership on organizational climate through teamwork were significant (B = 0.144, *p* < 0.01, 95% CI: 0.1252 to 0.1640).

Figure 1 shows the findings of the mediations model between leadership on the organizational climate through teamwork. These results support H1a.

According to the moderation model, the moderation effect of recognition on the relationship between quality of leadership and organizational climate was significant (B=0,009, *p* < 0.01, 95% CI: 0.002 to 0.0016). In this sense, recognition was a significant moderation effect on the organizational climate in workers with a low level of recognition (B = 0.2495, *p* < 0.01; 95% CI: 0.2209–0.2782), medium level (B = 0.2695, *p* < 0.01; 95% CI: 0.2437–0.2953) and high level (B = 0.2945, *p* < 0.01; 95% CI: 0.2615–0.3275). These results partially support H1b.

## 4. Discussion

This research expects to confirm the relationship determined in other investigations [2,11,20], that is, how the quality of leadership influences the climate perceived by workers in the organization. In addition, we intend to observe what effect other mediator and moderator variables have on this relationship, that is, if the teamwork affects the relationship between the leadership and the organizational climate and how the recognition of achievement moderates the relationship between the leadership and the organizational climate.

To the best of the authors’ knowledge, to make predictions about the effects of leadership quality on the organizational climate perceived by workers, we have considered the relationship between these variables and, once determined, we have investigated the effects that teamwork and recognition of achievement have on that relationship through mediation and moderation analysis.

In this study, the quality of perceived leadership is found in medium values, the organizational climate was determined in medium-high values, the perception of teamwork in medium-low levels, and the recognition of achievement in low levels. Likewise, the correlations between leadership, recognition, teamwork, and organizational climate have been high or very high.

The fact that recognition of achievement presents high levels of correlation with the organizational climate and with the perception of leadership quality means that managers who have greater recognition towards their collaborators improve the perception of the organizational climate.

The high correlation between recognition of achievement and leadership indicates that when recognition is low, the perception of leadership quality is low, and when recognition of achievement is high, the perception of leadership quality is also high.

In this study, teamwork shows a high correlation with the organizational climate and moderate correlations with the quality of leadership and recognition of achievement.

From the results obtained on mediation and moderation, we deduce that there is a direct and significant relationship between the perception of leadership quality and the organizational climate.

In research by numerous authors [11,12,20,27], leadership presented a high correlation with the organizational climate, so to improve the climate in the organization, it is necessary to implement a leadership style based on the people.

A leadership style based on the people requires leaders to develop social skills to ensure the respect and confidence of their followers, to lend their support and encouragement, and to facilitate opportunities for participation to followers [25]. The leader has to accept individual differences by making sure that interactions with followers are personalized [26].

Similarly, the quality of leadership has a direct and significant influence on the perception of the congruity of teamwork, that is, the better the perception of the quality of leadership, the greater the perception of the congruity of teamwork.

Obviously, leadership and teamwork cannot exist without each other. They have to be balanced, coordinated, and synergized for optimal organizational performance towards successful outcomes [23].

Teams are an essential component of successful organizations today, and building and motivating these teams are necessary pursuits to attain that success. Teams require continuous nurturing and interaction to maintain high performance throughout their temporary lives. Leadership must now concentrate on motivating and supporting teams using tools that were not previously considered, but have become crucial in a globalizing environment [23].

The hypothesis H1a, if the team work mediates the relationship between the leadership predictor variable and the organizational climate variable, according to the results of this study, we can affirm that the teamwork variable significantly exerts the expected mediating effect, so that the quality of leadership relationship with the organizational climate is strengthened when the quality of leadership is perceived by the work team, significantly improving the organizational climate.

Teamwork has a great impact on the organizational climate, because it is necessary that there be an alignment of team members for what is required and interaction of underlying aspects such as leadership [28] and the recognition of workers.

A good work climate facilitates the interrelation between the members of the work team, enables improvements individually and collectively, improves communication, generates high levels of performance, and enhances the qualities of the different team members. This demonstrates the close relationship that exists between the work climate and teamwork, as well as its impact on the achievement of results in terms of productivity, achievement of goals, and creation of culture [29], Likewise, the greater the degree of cohesion in the work team, the better the perception of the work climate [30].

The role of leaders and managers in the climate of work teams is determined by numerous factors on which they act and they can significantly influence the climate of the teams they lead [18]. 

Leaders and directors can contribute significantly to modeling the climate of the teams they lead. The leader’s information quality, the frequency of leader-member interaction, transformational leadership, and the variability, simplicity, and visibility of the behavior patterns of those responsible are important factors in this regard. Given that, the climate of the work teams can have an important influence on the results of the team [18].

In regard to hypothesis H1b, if the recognition of leadership achievement moderates the relationship between the leadership predictive variable and the criterion variable organizational climate, we can affirm that recognition of achievement does not change the relationship of leadership to the organizational climate and, therefore, there is no moderation of the recognition of achievement in that relationship.

Among the characteristics that positively influence the perception of the leader’s behavior is recognition as a necessary motivating factor for workers [31]. Among the qualities necessary for good leadership is the recognition of the achievements of workers as an innate condition of the leader.

According to Maslow’s theory of human motivation (1943), recognition is the fourth necessity in its hierarchical scale. Thus, those workers who have covered the first three needs will try to cover this fourth and if it does not occur, they will not be able to reach the fifth—the need for self-realization or personal growth—presenting employees with lower levels of motivation [32] and confidence who will not develop their full potential, which may lead to internal processes of frustration affecting the organizational climate. [8,28].

The recognition of achievement must come from the direct manager of the worker, but they can also receive it from the organization and the work environment, such as colleagues, clients, patients, students and others with whom are work-related, as is indicated in the research [19,33] There are plenty of ways for management to show employees the recognition they deserve. The two most underused words in any organization or company are the simple words, ‘Thank you’’ or “Good work”. While most employees certainly appreciate monetary awards for a job well done, many people merely seek recognition of their hard work. In order to develop an effective recognition program, however, the management should remain flexible in the methods of recognition, as different employees are motivated by different forms of recognition.

### 4.1. Limitations and Future Directions of Research

This work has some limitations. In the first place, as regards to the composition of the sample, there is a greater weight of specific geographical areas with respect to the rest and there is a greater representation of men than women. This fact may have an effect on the results obtained if it is considered that the sample also has a self-selection component. Second, the data is self-reported and it seems necessary to triangulate the data collection sources; and finally, the cross-sectional nature of the study limits the possibilities of establishing cause-effect relationships between the variables and hence, longitudinal research designs are proposed for the future.

For future research, it is proposed for inclusion in the study variables that can moderate significantly the relationship between leadership and organizational climate, such as companionship and social support, to check whether the strong ratio determined in this study between leadership quality and organizational climate is moderable and, therefore, if we have an alternative course of action that compensates for the low quality of leadership for the improvement of the organizational climate.

Additionally, it would be of interest as a future line of research, to replicate the study in an organization, company or study population in which the values of the recognition made by those responsible and the management were high, thus checking what effects this change had in the relationship between the variables.

### 4.2. Recommendations for Practice

The results of our study can be useful for companies that want to adapt and be part of the changes in the VUCA environment—Volatility (V), Uncertatinty (U), Complexity (C), and Ambiguity (A)—where we are currently and companies are developing [34]. The main challenge facing companies is to have the human resources capable of driving change [35], guided by an agile and operational transformational leadership that allows the development of the work team in a collaborative environment based on trust and commitment.

Leadership should facilitate the personal initiative and active participation of the members of the work team so that in being motivated, they push the necessary innovative ideas in the changing environment [36]. For this to happen, the leadership must recognize the individual contributions to achieve the challenges and objectives of the work team.

The digital transformation of companies as a lever for change in VUCA environments can also generate the dehumanization of work if companies do not accompany that change with person-oriented policies [37,38]. In changing environments, human resources can be very vulnerable if they are not guaranteed adequate levels of organizational climate, so the challenge of organizations is to enhance corporate well-being to improve the quality of life of employees and achieve higher levels of satisfaction, commitment, and a feeling of belonging to the organization [21].

## 5. Conclusions

In changing environments, motivated and capable human resources are required to drive change [36], to achieve this, the challenge of companies is to guarantee corporate well-being through adequate levels of organizational climate, to improve the quality of life of employees, and achieve higher levels of satisfaction, commitment, and retention of talent.

The quality of leadership and the recognition of achievement have a great impact on the organizational climate based on the perceptions caused by the leader, as they point out [2], which is explained by the strong correlation between the three variables. However, recognition of achievement does not moderate or change the relationship of leadership quality to the organizational climate, which can be explained by considering recognition of achievement as an innate or necessary condition of good leadership as they point out [31]. It is necessary that the organization define and establish its own person-centred leadership style that allows it to achieve its objectives as an organization in a work climate that facilitates the quality of life at work for the development of workers.

Teamwork is a way of organizing work that is well valued by companies for the achievement of their objectives, but to successfully develop teamwork, it is required that from the company that the necessary conditions be promoted and facilitated so that the created work groups are constituted and function as teams that work in a collaborative environment based on trust and commitment, and that the leadership exercised over the work team is perceived by the members of the team to be adequate. Therefore, the role of the leader is to encourage the development of the team that leads the members of the team. In this sense, recognition of achievement plays a fundamental role because by recognizing the team as a whole, the leader must ensure that each member with their individual differences feels valued for their contribution to the proper functioning of the work team.

In conclusion, the leadership is the trigger for a big job performance through the construction of a good organizational climate based on perceptions caused by the leader [2], by applying leadership styles based on the recognition of achievement and tteam work in a collaborative and trustful environment among team members whose individual differences enhance team results and help the success of the organization.

## Figures and Tables

**Figure 1 ijerph-17-00032-f001:**
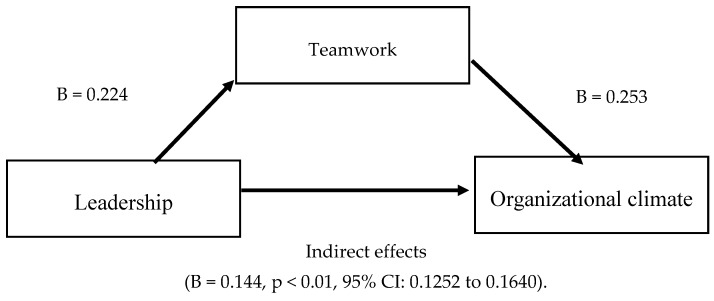
Conceptual diagram with direct and indirect effects of the mediation model.

**Table 1 ijerph-17-00032-t001:** Mean, Standard Deviation, Cronbach’s alpha, and Pearson correaltion between all dimensions.

	α	M	SD	1	2	3
Organizational climate	0.94	54.3	16.09			
Quality of leadership	0.90	57.9	22.96	0.830 **		
Recognition	0.73	51.9	22.37	0.803 **	0.703 **	
Teamwork	0.74	45.6	19.17	0.779 **	0.681 **	0.627 **

** *p* < 0.01.

**Table 2 ijerph-17-00032-t002:** Moderated-mediation analyses of the quality of leadership and recognition on the organizational climate mediating pathway for teamwork. Explained variable: organizational climate.

Variables	B	SE	t
Constant	16.37	0.958	17.08**
Quality of leadership	0.224	0.020	10.94 **
Teamwork	0.253	0.014	17.67 **
Recognition	0.198	0.022	8.69 **
Recognition * Quality of Leadership	0.009	0.003	2.65 **

Nota: B standarized coefficients; R^2^ = 0.831 (N = 1179), ** *p* < 0.01.

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
