# Peer review of "Quality of Leadership and Organizational Climate in a Sample of Spanish Workers. The Moderation and Mediation Effect of Recognition and Teamwork"

_ijerph, 2019, doi:10.3390/ijerph17010032_

Round 1

Reviewer 1 Report

Thank you very much for the opportunity to review this paper title: Quality of leadership and organizational climate in a sample of Spanish workers”. The moderation and mediation effect of recognition and teamwork.” It is an interesting topic. However, the overall manuscript is not acceptable standards for publication in any impact factor (SSCI) journal. The topic deserves further investigation, and there are a number of issues that need to be addressed before considering this paper for publication. In its current form, the manuscript is still in a very initial stage, pretty far from the quality standards of an academic paper. Before further process of this paper, I suggest need a major revision in this paper. I explain some of my reservations in detail below

Abstract: In the abstract, the statically techniques which authors use in this study is not explain. Moreover, is it suggesting that remove the heading of the in the abstract and improve the obstruct of your study? Introduction: In the introduction, the part provides a more rational background of the gap of the study and research questions. As well as explain your research questions in detail that what research you are going to do? Hypotheses Development: After the introduction, you have written a direct research methodology. I suggest you made new heading with the title of “Hypotheses development.” Under this heading explain rational background of the hypotheses development with the support of the previous literature. Methodology: The methodology of the study is not well-explained authors need a lot of improvement in this part of the study. For example, first subheading Participants and Procedure (2.1) is explaining the comprehensive procedure of research methodology, and seconding subheading title is Instruments (2.2) is explaining the measurements of the variables. It is not explained as research methodology explains in the social sciences studies. The validation of the questionnaire is doubtful. Please, provide more details, I want to see the questionnaire of the study can you please provide the questionnaire. I also suggest you break your methodology into four subheadings i.e. i. Research approach (explain in detail what kind of research approach authors are using in this study), ii-Questionnaire designing (Questionnaire design or instrument development and write in the detail how you develop your questionnaire and how do you test your questionnaire before data collection) iii-Participant and procedure (Explain in detail sampling techniques that you use in this study as well as the target samples and procedure of data collection)  and iv. variables measurements (Explain each variable from which study you adopt this variable and what was your sample questions as well as what is the Alpha value of each variable and what is the standardly accepted value). I have seen you have tried to write variable measurements under heading 2.2 Instruments, but it is not the research-oriented way to express the merriments of the variables. Results: The results section also very weak. I suggest first, we need to explain in detail validity and reliability of the study. Secondly, you also need to explain first direct effect of the independent and dependent variables as well as separately explain the mediation and moderation effects. Discussion: The discussion part must be explaining in the detail and interlinked with the results and previous literature. Moreover, you have written two to three lines paragraph in the methodology. It is a research paper, not an assignment. I suggest you follow the discussion style as is written in below mentioned paper.

Rasool, S.F.; Maqbool, R.; Samma, M.; Zhao*., Y.; Anjum, A. (2019). Positioning Depression as a Critical Factor in Creating a Toxic Workplace Environment for Diminishing Worker Productivity. Sustainability11 (9), 2589 (1-18).

Conclusion: The conclusion is feeble; it is not well explained. There are no concluding remarks. Key learnings, implications for policy and practice, limitations, and future research avenues are also missing. I think we will discuss this first then we will try to improve this section of the paper. Managerial implications: We must explain and rewrite the what level of managers as well as what kind of mangers (Functional managers) will apply this study at their profession and what will be the outcome of this implications

Reviewer 2 Report

Interesting work, without being innovative, but the authors showed caution in framing the constructs addressed. The work is interesting for the mastery of the study of the work context and the latent factors that workers may experience, regarding the psychological variables.

A revision of the text references will be necessary as they do not comply with the APA standards. Revise the entire text and correct the various points where the criteria in the works of this nature are not met.
The method is adequate for the presented objective, as well as the formulated hypotheses.

The applied scales, despite Cronbach's alpha, do not show the FIT indices for each scale. It was more robust, even though the Delphi method was applied.
The mediation and moderation models applied, although in table, could be illustrated in figure.

They can extend the practical applications of the results obtained to the real context. In addition to the cohesion of scientific results, practical applications become attractive to potential stakeholders in this field of study.

Please note that references in the bibliography do not in many cases comply with APA standards. Review and correct the bibliography.
Of the 33 references in the bibliography only 9 references are 5 years old or younger.

Round 2

Reviewer 1 Report

It is an interesting topic. However, the I reject this manuscript for publication. I explain some of my reason of rejection in detail below.

English: First of all, I need to point out that this paper suffers from a great amount of grammar and spelling mistakes, as well as typos and other formal flaws. In turn, this paper is partially non-understandable or potentially not in a way that you were intending. Introduction: Introduction is still very weak even I suggest to authors in first revision how to write introduction. Authors have not written more rational background of the gap of the study and research questions in this study. As well as they have not explained the research questions in the introduction. Hypotheses Development: The hypotheses development is very weak; it is pretty far from the quality standards of an academic paper. Methodology: The authors improve methodology of this study but still it is not up to the standard of this journal. Results: In first revision I suggest to the authors please explain in detail validity and reliability of the study. But authors are enable to test the validity and reliability test in the revised paper. Secondly, I also was suggested to explain the direct effect of the independent and dependent variables as well as separately explain the mediation and moderation effects. But authors were not explained separately direct and indirect effect of the variables. Discussion: The authors try to improve the discussion section but still a lot of issues. Discussion section of this paper is looking like a class assignment. Conclusion: The conclusion is still feeble.
